# Biochemical and Nutraceutical Characterization of Different Accessions of the Apricot (*Prunus armeniaca* L.)

Aurora Cirillo [1],*, Lucia De Luca [1], Luana Izzo [2], Marco Cepparulo [1], Giulia Graziani [2], Alberto Ritieni [2], Raffaele Romano [1] and Claudio Di Vaio [1]

[1] Department of Agricultural Sciences, University of Naples Federico II, Via Università 100, 80055 Portici, Italy
[2] Department of Pharmacy, University of Naples Federico II, Via Domenico Montesano 49, 80131 Naples, Italy
*   Correspondence: aurora.cirillo@unina.it

**Abstract:** The name "Vesuvian apricot" refers to numerous varieties originating from the same production area at the foot of Vesuvius and with a long tradition of cultivation. The importance of the apricot fruit and its derivatives on human health is known thanks to the presence of several secondary metabolites, many of them being active as antioxidants. This study aims to analyze 12 vesuvian apricot accessions by means of chemical-physical (fruits weights, firmness, TSS, TA, pH, and color fruits) and nutraceutical analyses (acid organic and sugar content, antioxidant activities, and polyphenol content). All the accessions analyzed can be defined as superior-quality apricots because they had TSS values $\geq 13$ and a pulp strength $\leq 1 \, \text{kg}/0.5 \, \text{cm}^2$. Another parameter used to express the quality of apricot fruits was TSS-F (soluble solids content-pulp hardness) which showed a value $\geq 12$. The total sugar content ranged from 260.40 mg/g d.w in 'VM' to 744.59 mg/g d.w in 'SC'. In all accessions analyzed the sugar content was in the following order: sucrose > glucose > fructose. The antioxidant activity showed a high variability between the different accessions. Our results show that the Vesuvian apricot accessions present a large range with different organoleptic characteristics and offer the possibility to choose according to consumer and processing preferences.

**Keywords:** Vesuvian apricot; acid organic content; sugar content; antioxidant activity; total polyphenols



## 1. Introduction

*Prunus armeniaca* L. is a species that is principally cultivated in the Mediterranean countries and in Asia Minor (Turkey, Uzbekistan, Iran, Algeria, Italy, Spain, France, and Greece), where many local cultivars are grown, representing more than 75% of the world production of apricots [1,2]. Apricot is particularly suitable for low-input growing agricultural systems as observed in southern Italy, where organic apricot production is spreading widely [2]. The total world production is approximately 2.6 million tons, with Uzbekistan and Turkey as the leading countries in the production of fresh apricot fruits (www.fao.org/faostat/en/#data/QC accessed on 10 November 2021) [3]. Italy is the fifth worldwide and the first European producer of apricots, with 222.690 tons (www.fao.org/faostat/en/#data/QC accessed on 10 November 2021). The most important area for apricot cultivation lies around Mount Vesuvius, in the Campania region, with a production of approximately 40%. [4]. Vesuvian apricots have superior qualitative characteristics that make them well suited for processing and as ingredients in traditional confectioneries and pastries [4]. The remaining apricot fruit production is distributed to local markets, as Vesuvian apricots become marketable later than fruit imported from other countries. The name "Vesuvian apricot" refers to a set of numerous varieties (over 40), all originating from the same area at the foot of Vesuvius. The most popular varieties are: Ceccona, Palummella, San Castrese, Vitillo, Pellecchiella, Monaco Bello and Portici [5]. The area devoted to apricot cultivation around Mount Vesuvius is gradually decreasing, mainly because of the urbanization of rural areas and the importation of apricots is more

widespread. These varieties often have a regional diffusion and are linked to specific geographic areas and traditions and, therefore, they can be considered landraces today [6,7]. Apricot landraces have lost competitiveness and are mainly cultivated in small farms [8].

Apricot fruit is highly appreciated by consumers and is among the most important fruit species grown in the world. It is considered by many to be among the most delicious fruits, and a good balance of sugars and acids and a strong apricot aroma are major determinants of exceptional fruit quality [9]. In recent years, research has aimed at enhancing the qualitative parameters of apricot, and specific breeding programs have focused on the improvement of apricot flavor through determination of the roles played by principal acids and sugars, aiming to produce new cultivars with better traits such as good flesh taste, aroma, and firmness, high sugar content, big size and attractive fruit color, extensive harvesting period, and resistance to Sharka disease, as all local European cultivars are susceptible [9,10]. Fruit quality is fundamental for the acceptance of apricot cultivars by consumers [11], and this concept depends on sensory properties (appearance, texture, taste and aroma), nutritional values, chemical compounds, and mechanical and functional properties [12]. So, new apricot cultivars must be characterized by fruit quality attributes that satisfy consumers. The sensorial properties of apricot fruits are principally influenced by the sugars, organic acids, and volatile compound contents, color, size, texture [11], firmness, attractiveness, and taste [1,13]. Furthermore, the importance of apricot and by-products for human health is well ascertained. In fact, it can exert different positive effects such as against cancer, cardiovascular disease, atherosclerosis, and aging-related diseases; it can protect the kidneys and liver [14]. Apricot contains several secondary metabolites [15,16], many of them being active as antioxidants [17]. Polyphenols and carotenoids represent the most abundant classes of phytochemicals contained in this fruit [18]; such molecules also play a primary role in the color and taste of fruit [19,20]. Many studies have been carried out to characterize the quality, composition and biochemistry [9,10,13,21] of apricots belonging to the list of new cultivars, whereas very few have been conducted on the quality attributes of autochthonous/traditional varieties of the Campania region. This is surprising because, although apricot cultivation in Italy and Campania has been subjected to a significant cultivar turnover, traditional varieties are still widely cultivated and appreciated [22].

In a broad varietal scenario, it is important to identify the apricot varieties with the best production and quality characteristics capable of satisfying the needs of producers and consumers. In this regard, our study aims to describe 12 apricot accessions belonging to the Vesuvian apricot germplasm of the Campania region from a biochemical and nutraceutical point of view, with the aim of selecting among them the best both in terms of organoleptic quality and health-promoting effects.

## 2. Materials and Methods

### 2.1. Plant Material

The experimental trial was conducted on different apricot genotypes belonging to the Germplasm collection realized at "Improsta" Regional Experimental Farm, in Eboli (Southern Italy) (40°33′29″ N; 14°58′28″ E, 15 m a.s.l.). The research activities in the field were carried out in 2022, starting on June 24th and ending on July 7th. This study was performed on 12 apricot accessions, namely 'Vincenzo e Maria', 'Fracasso', 'Piciona', 'Boccuccia di Eboli', 'Scassulillo grande', 'Palummella',' Paolona', 'Puscia', 'Panzona', 'Pellecchiella', 'Portici 2', and 'Vitillo'. Individual plants of all apricot accessions were thirteen years old, grafted on Myrabolan 29C (*Prunus cerasifera*) and vase trained (4 m × 4 m spaced). The experiment set up was organized as a completely randomized block design, with five trees/replicates per genotype and the analyses were carried out on 100 fruits divided for the various analyses. In Table 1, the abbreviations of accessions and time harvested are reported.

**Table 1.** Harvesting period and abbreviations of the 12 Vesuvian apricot accessions analyzed.

| Accessions | Harvesting Time | Abbreviations |
|---|---|---|
| Boccuccia di Eboli | 30 June 2022 | BE |
| Fracasso | 7 July 2022 | FR |
| Palummella | 24 June 2022 | PAL |
| Panzona | 7 July 2022 | PAN |
| Paolona | 30 June 2022 | PAO |
| Pellecchiella | 27 June 2022 | PE |
| Piciona | 24 June 2022 | PI |
| Portici 2 | 3o June 022 | PO |
| Puscia | 30 June 2022 | PU |
| Scassulillo grande | 30 June 2022 | SC |
| Vitillo | 27 June 2022 | VI |
| Vincenzo e Maria | 24 June 2022 | VM |

*2.2. Physico-Chemical Analysis*

Each sample consisted of 15 apricot fruits harvested at consumer maturity, which showed a quite low pulp firmness; in fact, apricot softening is a crucial parameter along the whole post-harvest supply chain, one that is highly influenced by the cultivar, storage conditions and duration [23]. The parameters analyzed were weight (g), pulp firmness (F) (kg/0.5 cm$^2$), epicarp coloring (using color coordinates L*, a*, b* CIELAB), total soluble solids content (TSS), pH, and titratable acidity (TA). The weight was determined with an electronic digital balance (Precisa Instruments AG, model XB220A, Dietikon, Switzerland), while the pulp firmness was determined with an EFFEGI manual penetrometer with an 0.5 cm$^2$ tip on two sides opposite the fruit. The color attributes of the epicarp were measured using a Minolta CR-400 Colorimeter (Konica Minolta, Inc., Osaka, Japan) to delineate chromaticity values L* (lightness), a* (green to red), and b* (blue to yellow). The measuring was repeated two times in different points of the fruit. Subsequently, the chroma parameter [23] was calculated using the formula:

$$(a)\ (a^{*2} + b^{*2})^{1/2}.$$

Total soluble solid (TSS) content, expressed as °Brix, was recorded using a digital refractometer (Atago, model PR-101a, Tokyo, Japan), and the pH was measured with a digital pH meter (Crison Instruments, model GLP 21, Barcelona, Spain). The total acidity was determined with an acid–base titration, the solution was titrated with 0.1 N sodium hydroxide standard solution and expressed as g citric acid 100 mL$^{-1}$. Finally, the TSS/TA and TSS-F ratios were calculated.

*2.3. Determination of Sugar*

The extraction of individual sugar was performed following the method proposed by İmrak et al. [24] with modifications. Briefly, 1 g of the freeze-dried sample was mixed with 10 mL of deionized water. The solution was centrifuged at 6500 rpm for 10 min. Subsequently, the aqueous extract was filtered with a PES filter of 0.45 μm and injected (20 μL) in HPLC. The HPLC analysis was performed using a HPLC apparatus (Agilent 1100; Agilent Technologies Inc., Santa Clara, CA, USA) equipped with a refractive index detector (G1362A). The HPLC method proposed by Karatas [25] was used. The isocratic mobile phase was water/acetonitrile (25:75, *v/v*), the flow rate was set to 1.7 mL min$^{-1}$ and a ZORBAX carbohydrate NH$_2$ column (4.6 × 250 mm, 5 μm; Agilent Technologies Inc.) was used. To calculate the concentration of sugars, calibration curves with glucose, galactose and lactose standard solutions (5000–30,000 ppm) were constructed. The results were expressed as mg/g of dry weight (d.w.).

### 2.4. Determination of Organic Acids

The organic acid concentration was evaluated by HPLC analysis. The sample was extracted following the method proposed by Fan et al. [26], with modifications. Briefly, 0.5 g of freeze-dried apricots was mixed with 20 mL of deionized water. The solution was centrifuged at 6500 for 20 min. Finally, the aqueous extract was filtered with a 0.45 μm PES filter and injected in HPLC apparatus. The organic acid contents were determined by injecting a 20 μL supernatant into an HPLC (Agilent 1100; Agilent Technologies Inc., Santa Clara, CA, USA) equipped with a diode array detector, following the method proposed by Manzo et al. [27].

The UV detection was performed at 210 nm. The isocratic mobile phase was water acidified with orthophosphoric acid (pH 2.1) and the flow rate was set to 0.8 mL min$^{-1}$. A reversed-phase column, S5 ODS2 column (4.6 × 250 mm, 5 μm; Agilent Technologies Inc.), was used. To calculate the organic acid concentration, calibration curves were constructed using organic acid standards (ascorbic acid, citric acid, fumaric acid, malic acid, shikimic acid, succinic acid and tartaric acid). The range of linearity was 25–500 ppm for ascorbic and succinic acid, 10–500 ppm for shikimic acid and 50–500 pm for all other standards. The results were expressed as g/kg of dry weight (d.w.).

### 2.5. Chemicals and Reagents and Polyphenolic Extraction

Methanol, ethanol, formic acid, and acetonitrile of HPLC grade were acquired from Merck (Darmstadt, Germany). Standards of gallic acid, (±)-6-hydroxy-2,5,7,8-tetramethyl-chromane-2-carboxylic acid, commonly namely Trolox, 2,2-azinobis (3-ethylbenzothiazoline-6-sulphonic acid) diammonium salt (ABTS), ferrous chloride, 1,1-diphenyl-2-picrylhydrazyl (DPPH), 2,4,6-tris(2-pyridyl)-1,3,5-triazine (TPTZ), and Folin–Ciocalteu reagent were purchased from Sigma Aldrich (Milan, Italy). All other chemicals and reagents were of analytical grade. Polyphenols were extracted according to the procedure reported by Iglesias-Carres et al. [28] with some modifications. Briefly, 1 g of the freeze-dried samples was introduced into a 50 mL Falcon tube and extracted with 5 mL of a MeOH:H$_2$O (80:20, *v/v*) mixture with 0.1% FA. The samples were vortexed (ZX3; VEPL Scientific, Usmate, Italy) for three min, sonicated (LBS 1; Zetalab srl, Padua, Italy) for ten minutes, and then agitated for an additional ten minutes using a digital orbital shaker (SKO-D XL ARGOlab, Arezzo, Italy). The mixture was then centrifuged at 5000 rpm for 5 min at 4 °C. The supernatant was collected and then the pellet extracts at another time with 5 mL of a MeOH:H$_2$O (80:20, *v/v*) mixture with 0.1% FA. The supernatant collected with the other aliquot was filtrated through a 0.22 μm filter and diluted ten times for the UHPLC Q-Orbitrap HRMS analysis [29].

### 2.6. UHPLC-Q-Orbitrap HRMS Analysis

The polyphenol profile was determined by using an Ultra High-Pressure Liquid Chromatograph (UHPLC, Dionex UltiMate 3000, Thermo Fisher Scientific, Waltham, MA, USA) equipped with a degassing system, a quaternary UHPLC pump working at 1250 bar, and an autosampler device. Chromatographic separation was performed with a thermostatic (T = 25 °C) Kinetex 1.7 μm F5 (50 × 2.1 mm, Phenomenex, Torrance, CA, USA) column. The mobile phase consisted of water (A) and methanol (B), both containing 0.1% FA. The injection volume was 5 μL. The total run time was 13 min, and the flow rate was set to 0.5 mL/min. The gradient elution protocol consisted of an initial 0% B, an increase to 40% B in 1 min, a rise to 80% B in 1 min, and an increase to 100% B in 3 min. The gradient was maintained at 100% B for 4 min, then lowered to 0% B for 2 min, and then kept at 0% for another 2 min to allow for column re-equilibration. The mass spectrometer was operated in both the positive and negative ion modes by setting 2 scan events: full ion MS and all ion fragmentation (AIF). The following settings were used in the full MS mode: a resolution power of 70,000 full width at half maximum (FWHM) (defined for $m/z$ 200), an automatic gain control (AGC) target 1 × 106, a scan range of 80–1200 $m/z$, an automatic gain control (AGC) target of 1 × 106, an injection time set to 200 ms and a scan rate set to 2 scan/s.

The following ion source characteristics were used: a spray voltage of 3.5 kV, a capillary temperature of 320 °C, a S-lens RF level of 60, a sheath gas pressure of 18, an auxiliary gas of 3, and a heater temperature of 350 °C for auxiliary gas. The following parameters were configured for the AIF scan event in both the positive and negative modes: ACG target: $1 \times 105$; scan range: 80–120 $m/z$; isolation window: 5.0 $m/z$; retention time: 30 s; mass resolving power: 17,500 FWHM; maximum injection time: 200 ms; scan time: 0.10 s. The collision energy was varied in the range from 10 to 60 eV to obtain representative product ion spectra. Identification and confirmation were carried out at a mass tolerance of 5 ppm for the molecular ion and for both fragments. Data analysis and processing were performed using Xcalibur software, v. 3.1.66.10 (Xcalibur, Thermo Fisher Scientific, Waltham, MA, USA).

### 2.7. Determination of Total Phenolic Content

According to the method described by Izzo et al. [29], the Folin–Ciocalteu method was employed to determine the total phenolic content. In short, 125 μL of the extract was mixed with 500 μL of deionized water and 125 μL of the Folin–Ciocalteu reagent 2 N. After mixing, the tube was kept in the dark for 6 min. Then, 1 mL of deionized water and 1.25 mL of a 7.5% sodium carbonate solution were added. The reaction mixture was kept in the dark for 90 min. Finally, a spectrophotometer was used to detect the absorbance at 760 nm. Data were presented as mg of gallic acid equivalents (GAE)/g of dry weight sample.

### 2.8. Antioxidant Activity

#### 2.8.1. Free Radical-Scavenging Assay (DPPH)

The method suggested by Brand-Williams et al. [30], with modifications, was used to determine the total free radical-scavenging activity of the samples. Briefly, 4.0 mg of DPPH was dissolved in 10 mL of MeOH, and the solution was then diluted to reach an absorbance of 0.90 ($\pm$0.02) at 517 nm. Volumes of 1 mL of the working solution and 200 μL of sample extract were combined to perform the experiment. Results were expressed as mmol Trolox Equivalents/kg of dry weight sample.

#### 2.8.2. Radical Cation Scavenging Assay (ABTS)

The method described by Re et al. [31] was used to measure the free radical-scavenging activity. Briefly, ABTS diammonium salt was dissolved with deionized water to reach a concentration of 7 mM to whom 44 μL of solution of potassium persulfate (2.45 mM) were added. The solution was stored at room temperature in the dark for 16 h. After that, ethanol was used to dilute the ABTS•+ solution until it had an absorbance value of 0.70 ($\pm$0.02) at 734 nm. Thereafter, 0.1 mL of the appropriately diluted sample was added to 1 mL of ABTS•+ solution. The absorbance was measured at 734 nm after a 2.5 min wait. Results were expressed as millimoles of Trolox Equivalents/kg of dry weight sample.

#### 2.8.3. Ferric Reducing Antioxidant Power (FRAP)

The ferric reducing antioxidant power (FRAP) of samples was determined spectrophotometrically following the procedure reported in a previous work [32]. The TPTZ solution (10 mM), ferric chloride solution (20 mM), and acetate buffer (0.3 M; pH 3.6) were combined in a ratio of 1:1:10 ($v/v/v$) to create the FRAP reagent. The assay was performed by using freshly prepared working FRAP reagent. Shortly, 150 μL of the diluted sample was added to 2850 μL of FRAP reagent. The value of absorbance was registered after 4 min at 593 nm. Results were expressed as mmol Trolox Equivalents/kg of dry weight sample.

### 2.9. Statistical Analysis

All data were subjected to analysis of variance (ANOVA). Duncan's multiple range test (DMRT) was performed for means separation of each of the measured variables at $p = 0.05$. A principal component analysis (PCA) was executed on TSS, TA, pH, organic acids and sugar content, antioxidant activity (FRAP, DPPH, and ABTS) and total polyphenols. R has

been calculated for relationship between the total phenolic content and antioxidant activity. The statistical package XLStat Version 2013 (New York, NY, USA) was implemented for all the analyses.

## 3. Results and Discussion

### 3.1. Physico-Chemical Analysis

The quality indices of the apricot accessions including fruit firmness, total soluble solids (TSS), titratable acidity (TA), and pH identified the fruits used in this study as ripe and ready-to-eat apricots. Figure 1 shows the photos of the 12 Vesuvian apricot accessions analyzed in this study; the fruit qualitative traits, reported in Table 2, indicated significant differences in the qualitative parameters of different accessions.

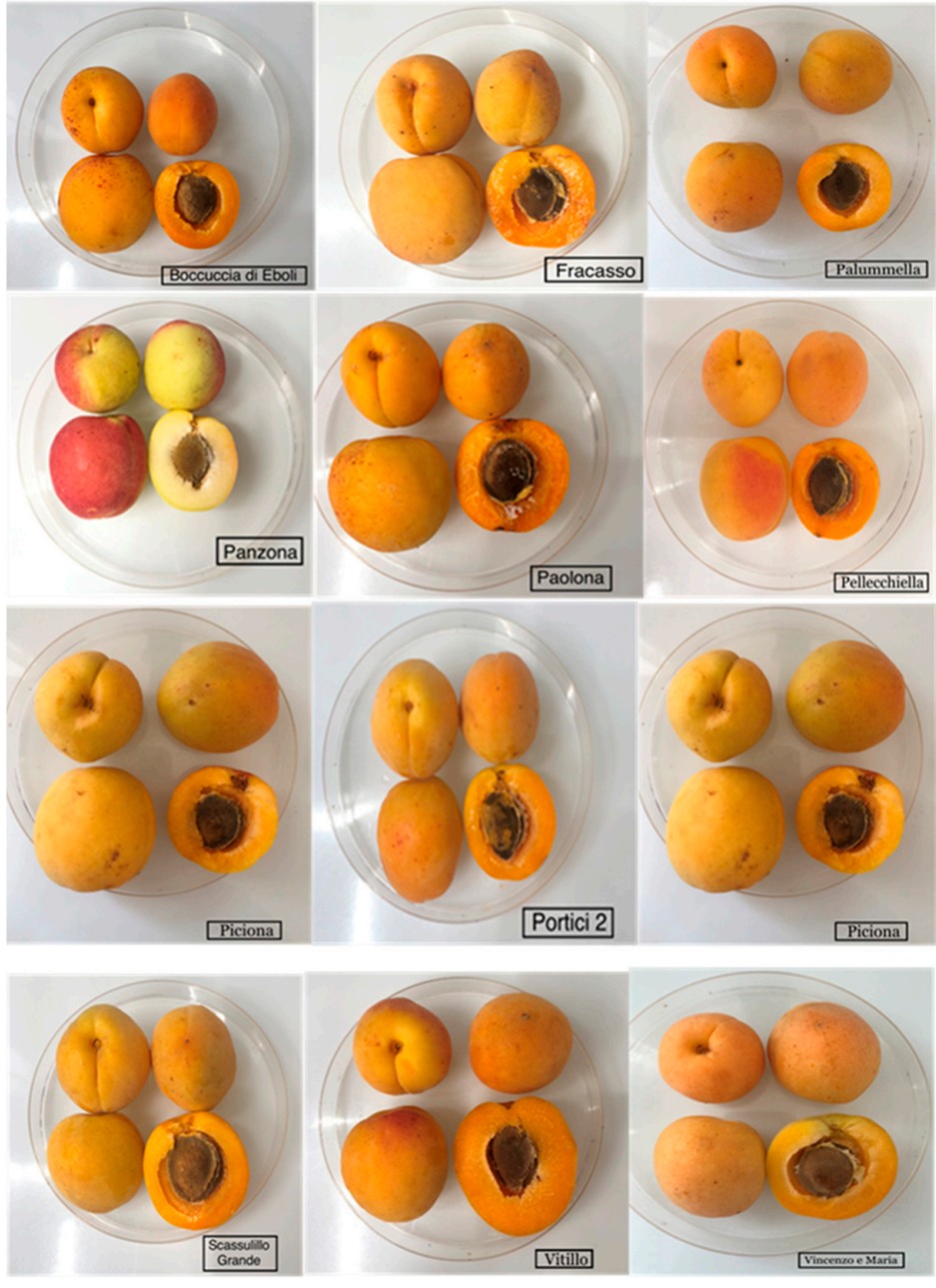

**Figure 1.** Vesuvian apricot accessions analyzed in this study.

**Table 2.** Fruit weight (g), pulp firmness (F) (kg/cm$^2$), total soluble solids (TSS), TSS-F, titratable acidity (TA), TSS/TA and pH at harvest for 12 of the different accessions of Vesuvian apricot.

| Accessions | Fruit Weight | Firmness (F) | TSS (°brix) | TSS-F | TA | TSS/TA | pH |
|---|---|---|---|---|---|---|---|
| | (g) | (kg/0.5 cm$^2$) | (°brix) | | (g/L Citric Acid) | | |
| BE | 36.98 ± 4.55 e | 1.24 ± 0.56 c | 16.53 ± 0.25 bcd | 15.29 ± 0.28 bc | 16.37 ± 1.36 b | 1.31 ± 0.12 cd | 3.71 ± 0.05 bc |
| FR | 60.37 ± 10.45 bc | 1.25 ± 038 c | 16.43 ± 0.31 cde | 15.19 ± 0.27 bc | 10.17 ± 0.71 fg | 0.62 ± 0.03 e | 4.16 ± 0.06 ab |
| PAL | 45.10 ± 2.58 d | 1.97 ± 1.02 b | 13.80 ± 0.26 fg | 11.83 ± 0.69 f | 22.17 ± 1.29 a | 1.60 ± 0.38 c | 3.22 ± 0.12 d |
| PAN | 44.47 ± 7.21 d | 2.97 ± 1.28 a | 17.00 ± 0.1 bc | 14.03 ± 0.72 de | 8.47 ± 0.21 gh | 1.07 ± 0.12 d | 4.20 ± 0.04 a |
| PAO | 60.17 ± 10.64 bc | 1.13 ± 0.63 c | 16.67 ± 0.23 bc | 15.54 ± 0.37 b | 6.90 ± 1.15 h | 1.31 ± 0.22 c | 4.16 ± 0.28 ab |
| PE | 42.37 ± 3.78 de | 0.84 ± 0.38 c | 14.37 ± 0.81 f | 13.52 ± 0.68 e | 11.07 ± 1.12 ef | 1.12 ± 0.02 d | 3.79 ± 0.37 abc |
| PI | 64.95 ± 8.25 b | 0.84 ± 0.62 c | 16.27 ± 0.49 cde | 15.43 ± 0.58 b | 10.50 ± 2.10 efg | 1.06 ± 0.08 d | 3.85 ± 0.31 abc |
| PO | 47.79 ± 6.36 d | 1.20 ± 0.44 c | 17.20 ± 0.52 b | 16.00 ± 0.58 b | 15.40 ± 0.40 bc | 1.38 ± 0.20 c | 3.46 ± 0.23 cd |
| PU | 56.97 ± 8.35 c | 2.10 ± 1.14 b | 15.83 ± 0.15 de | 13.73 ± 0.24 de | 15.03 ± 0.90 bc | 2.46 ± 0.41 a | 3.76 ± 0.15 abc |
| SC | 66.65 ± 10.36 b | 1.01 ± 0.60 c | 18.17 ± 0.40 a | 17.15 ± 0.35 a | 13.37 ± 0.35 a | 1.01 ± 0.07 d | 3.67 ± 0.42 c |
| VI | 88.33 ± 15.33 a | 1.99 ± 0.52 b | 13.33 ± 0.12 g | 11.35 ± 0.15 f | 12.53 ± 1.38 de | 1.62 ± 0.13 c | 3.72 ± 0.17 bc |
| VM | 57.35 ± 10.60 c | 1.21 ± 0.57 c | 15.73 ± 0.49 e | 14.52 ± 0.45 cd | 12.10 ± 0.82 def | 2.01 ± 0.06 b | 3.71 ± 0.28 bc |
| Means | 55.96 ± 8.21 | 1.48 ± 0.68 | 15.94 ± 0.34 | 14.46 ± 0.45 | 12.84 ± 1.12 | 1.38 ± 0.15 | 3.78 ± 0.21 |
| Significance | *** | *** | *** | *** | *** | *** | *** |

All the data are expressed as the mean ± SD (standard deviation). The same letter indicates not significant differences according to Duncan's multiple range test ($p < 0.05$). Level of significance per the ANOVA is indicated as *** ($p < 0.001$).

The analysis of fruit weight showed that 'VI' was the accession with the greatest weight (88.33 g), while the accession with the lowest weight is the 'BE' (36.98 g) and 'PE' (42.37 g). Our results agree with Di Vaio et al. [22], who analyzed three different Campania accessions (Ceccona, Vitillo and Pellecchiella), and "Vitillo" showed the greatest weight of 72.93 g. The same study reported fruit weight values of the "Pellecchiella" accession grown in the National Park of the Vesuvius area approximately 60% higher than the same accession analyzed in the present study (68.11 g vs. 42.37 g). The ANOVA test reported significant statistical differences between the weight of the various accessions which reported an average weight of approximately 55.96 g, this difference is due to the variability of the different genotypes analyzed, all grown under the same environmental conditions. Souty et al. [33] proposed the size, color, firmness, resistance to proposed manipulation, taste, aroma, and texture as the fundamental quality attributes in apricot fruit. In our study, pulp firmness ranged from 0.84 kg/0.5 cm$^2$ (for the 'PE' and 'PI' accession) to 2.97 kg/0.5 cm$^2$ ('PAN' accession), with a general average in the "Vesuvian apricots" of approximately 1.48 kg/0.5 cm$^2$, these values confirm the suitability of apricot fruit for fresh consumption. Our results highlighted lower pulp firmness values than those reported in the bibliography; in fact, Caliskan et al. [34] reported fruit firmness between 1.1 and 4.8 kg/cm$^2$ among early matured fourteen apricot cultivars grown in the Mediterranean region and harvested in May in Turkey. Gurrieri et al. [1] reported that firmness, attractiveness, and taste are the principal parameters affecting apricot fruit quality.

The TSS results (°brix) reported higher values in the 'SC' accession (18.17 °brix) and lower values in 'VI' (13.33 °brix), thus recording greater TSS of approximately 36.30%. The TSS results recorded in our study are similar to those reported by Di Vaio et al. [22] on 'Vitillo', which showed values of approximately 12.20 °brix and by Leccese et al. [2], with a value of approximately 12.10 °brix on the same accession. Juice acidity varied greatly among accessions, showing a range of 22.17 g/L of malic acid, for 'PAL' accession, and 6.90 g/L of malic acid for 'PAO', with a general average of 12.84 gL$^{-1}$ of malic acid. Our acidity results are similar to those reported by Lo Bianco et al. [9], who reported on 16 apricot cultivars in the northern coast of Sicily, with mean values of 11.3 gL$^{-1}$. Leccese et al. [35], in a study carried out on 18 apricot genotypes of the Italian and international germplasm, reported a TSS range of between 12 and 16 °Brix and TA ranged from 0.60 to 2.28% malic acid. The soluble solid content and the titratable acid are the most important factors affecting the taste of apricot. Ali et al. [36] showed that on apricot cultivars grown in Northern Areas of Pakistan had a total soluble solids range from 12.67 to 20.00 °Brix, while the variation in acid content in terms of malic acid on a fresh weight basis ranged between 4.5 and 8.6 of malic acid. Vesuvian apricots showed a higher quantity of soluble solids than other

apricot cultivars; in fact, Melgarejo et al. [37], in a study conducted on apricots grown in southeastern Spain, reported TSS values between 9.5 and 13 °brix.

The composition of sugars and organic acids, as well as the sugar–acid ratio, may influence the taste of apricots [38]. The TSS/TA ratio, an index which is used to determine fruit quality, was 0.62 for 'FR' and 2.46 for 'PU'. Compared to other fruit species, apricots are characterized by high sugar–acid ratio. Therefore, apricots are ideal for processing into juices, wine, and jams. Higher TSS/TA ratios correlate well with a higher eating quality [39]. Vesuvian apricot accessions present a choice of apricots with flavors from sour to sweet and offer the possibility to choose according to consumer and processing preferences. The pH showed results between 3.67 and 4.20, thus reporting higher values than the cultivars analyzed by Lo Bianco et al. [9], who reported a range from 2.2 to 3.6, with Ninfa (cv. early) and Orange Red (cv. intermediate) showing above-average values and Palummella, Goldrich, Alba and Bulida showing below-average values. It is reported that it is possible to define superior-quality apricots as fruits that have TSS values $\geq$ 13 and a pulp strength $\leq$ 1 kg\0.5 cm$^2$. Another parameter used to express the quality of apricot fruits is TSS-F (soluble solids content-pulp hardness), which must have a value $\geq$ 12 [40]. All the accessions analyzed show high values of TSS-F, with higher values in 'SC' (17.15) and lower values in the accession 'VI' (11.35). Overall, our results confirm that the analyzed Vesuvian apricots present excellent-quality fruits.

Table 3 shows the L*, a* and b* peel color coordinates of the 12 Vesuvian apricots. All peel color results indicated statistically significant differences among genotypes per $p < 0.05$ of L*, a* and b* values. Color is among the most important external factors of fruit quality, as the appearance of the fruit greatly influences consumers [11]. The study of the variability of apricot color is important for quality control to determine appropriate maturity, processing, consumption and selection of the best cultivar intended to be marketed. Color has a significant impact on consumer perception of apricot quality, especially regarding fruit attractiveness [41]. L* defines the lightness, and a* and b* define the red greenness and blue yellowness, respectively.

**Table 3.** Color parameters of peel (L*, a*, b*) and chroma (a*$^2$ $\times$ b*$^2$)$^{1/2}$ at harvest of the 12 accessions of Vesuvian apricot.

| Accessions | L* | a* | b* | Chroma |
|---|---|---|---|---|
| BE | 51.88 ± 4.36 cd | 28.62 ± 13.19 b | 40.37 ± 4.14 bc | 41.51 ± 6.88 e |
| FR | 54.39 ± 3.44 bc | 21.71 ± 9.97 b | 41.97 ± 3.67 abc | 52.28 ± 7.68 c |
| PAL | 52.90 ± 2.78 bcd | 20.73 ± 6.33 b | 41.38 ± 3.51 abc | 51.56 ± 5.73 c |
| PAN | 61.18 ± 14.09 a | 2.46 ± 47.39 c | 44.46 ± 17.49 a | 56.56 ± 8.52 b |
| PAO | 55.71 ± 4.81 b | 43.81 ± 15.39 a | 44. 17 ± 5.27 ab | 46.92 ± 8.41 d |
| PE | 52.23 ± 4.28 cd | 39.73 ± 11.13 a | 39.46 ± 3.75 | 49.39 ± 6.45 cd |
| PI | 59.30 ± 4.69 a | 24.31 ± 11.80 b | 45.04 ± 6.12 a | 46.56 ± 3.62 d |
| PO | 50.39 ± 5.77 d | 27.59 ± 13.46 b | 35.93 ± 6.66 d | 63.23 ± 11.51 a |
| PU | 59.03 ± 5.30 a | 20.43 ± 12.78 b | 43.15 ± 6.65 abc | 50.67 ± 8.24 cd |
| SC | 52.93 ± 3.29 bcd | 19.87 ± 6.43 b | 41.65 ± 3.32 abc | 48.21 ± 4.40 cd |
| VI | 53.16 ± 3.19 bcd | 30.23 ± 9.26 b | 41.05 ± 2.82 abc | 65.17 ± 14.79 a |
| VM | 53.04 ± 5.36 bcd | 22.00 ± 11.29 b | 33.79 ± 4.53 d | 46.75 ± 2.61 d |
| Means | 54.68 ± 5.11 | 25.12 ± 14.04 | 41.03 ± 5.66 | 51.57 ± 7.40 |
| Significance | *** | *** | *** | *** |

All the data are expressed as the mean ± SD (standard deviation). The same letter indicates not significant differences according to Duncan's multiple range test ($p < 0.05$). Level of significance per the ANOVA is indicated as *** ($p < 0.001$).

In 'PE', the peel color showed a more orange color, with a higher a* value (39.73) and lower L* value (52.23); conversely, 'PAN' showed a whiter skin color, with lower a* values (2.46) and higher L* values (61.18). Higher chroma values were reported in 'VM' and 'PU' with 65.17 and 63.23, respectively. It is reported that carotenoid content in apricot fruit has been shown to correlate well with skin and flesh color, with apricots with orange-colored flesh containing higher levels of carotenoids than white-colored flesh [33].

### 3.2. Sugar and Acid Content

In Table 4, the principal sugars content is shown.

**Table 4.** Sugar content (mg/g d.w) of the 12 accessions of Vesuvian apricot.

| Accessions | Fructose | Glucose | Sucrose | Total Sugar |
|---|---|---|---|---|
| | mg/g d.w | | | |
| BE | 11.97 ± 1.49 fg | 130.81 ± 0.16 g | 193.25 ± 6.48 g | 336.03 ± 7.80 f |
| FR | 42.62 ± 3.27 a | 190.49 ± 3.33 cd | 342.59 ± 3.17 c | 575.69 ± 6.77 b |
| PAL | 43.64 ± 4.25 a | 211.83 ± 14.44 b | 270.11 ± 10.72 f | 525.58 ± 22.91 c |
| PAN | 17.47 ± 5.41 e | 243.04 ± 1.79 a | 328.82 ± 5.43 cd | 589.33 ± 6.81 b |
| PAO | 30.25 ± 6.67 c | 197.52 ± 8.26 bc | 373.83 ± 6.17 b | 601.60 ± 15.10 b |
| PE | 35.40 ± 7.75 b | 205.69 ± 9.79 bc | 292.52 ± 5.55 ef | 533.61 ± 8.00 c |
| PI | 14.18 ± 8.81 ef | 178.18 ± 5.89 de | 342.21 ± 2.49 c | 534.57 ± 10.20 c |
| PO | 21.74 ± 9.52 d | 112.97 ± 2.74 h | 283.56 ± 0.79 f | 418.26 ± 0.43 e |
| PU | 36.37 ± 10.16 b | 150.97 ± 22.22 f | 293.48 ± 42.49 ef | 480.81 ± 69.88 d |
| SC | 28.14 ± 11.01 c | 163.20 ± 2.60 ef | 553.26 ± 0.98 a | 744.59 ± 3.58 a |
| VI | 23.51 ± 12.73 d | 239.23 ± 8.12 a | 315.50 ± 11.96 de | 578.23 ± 19.35 b |
| VM | 9.42 ± 2.45 g | 59.73 ± 1.91 i | 191.25 ± 2.91 g | 260.40 ± 0.55 g |
| Means | 26.23 ± 1.54 | 173.64 ± 6.77 | 315.03 ± 8.26 | 514.89 ± 14.28 |
| Significance | *** | *** | *** | *** |

All the data are expressed as the mean ± SD (standard deviation). The same letter indicates not significant differences according to Duncan's multiple range test ($p < 0.05$). Level of significance per the ANOVA is indicated as *** ($p < 0.001$).

The total sugar content ranged from 260.40 mg/g d.w in 'VM' to 744.59 mg/g d.w in 'SC', and the latter showed the highest TSS value (Table 2). In all accessions analyzed, the sugar content showed the following order: sucrose > glucose > fructose, confirming the order reported in previous works [26,42–44]. Sucrose was the most concentrated sugar, ranging between 191.25 mg/g d.w in 'VM' and 553.26 mg/g d.w in 'SC', and the content of glucose ranged between 59.73 mg/g d.w in 'VM' and 243.04 mg/g d.w in 'PAN', while the fructose content ranged between 9.42 mg/g d.w in VM and 43.64 mg/g d.w in 'PAL'. The content of both sucrose and glucose was higher compared to the range reported by Naryal et al. [42], who showed a range between 67.9 mg/g and 366.5 mg/g, and 37.3 mg/g to 152.9 mg/g, for sucrose and glucose, respectively, while the fructose content was lower for BE, PAN, PI, PO, VI and MI with respect to the minimum range reported by Naryal et al. [42], who showed a range from 27.6 mg/g to 155.3 mg/g, depending on the region. The results of sucrose and glucose concentration were similar to Fan et al. [26], who showed sucrose and glucose concentrations that ranged between 292.00 mg/g and 995.00 mg/g, and 82 mg/g and 303 mg/g, respectively, while the fructose content was lower compared to Fan et al. [27], who showed a concentration that ranged between 44 mg/g and 173 mg/g. On the other hand, Drogoudi et al. [10] showed sucrose, glucose and fructose concentrations that ranged between 284.00 mg/g, a higher concentration compared to sucrose content of BE and VM, and 449.00 mg/g, a lower concentration compared to SC; they found a range of glucose concentration from 53 mg/g, higher compared to VM, to 151 mg/g, a lower concentration compared to FR, PAL, PAN, PAO, PE, PI, VI, BE and a higher concentration compared to PO and VM, and a range of fructose concentration from 11 mg/g to 46 mg/g with a similar trend compared to our results, except for VM that showed a lower concentration (9.42 mg/g d.w). The concentration of sugar depended on the analyzed cultivar. Vega-Galvez et al. [45] showed a concentration of sucrose of 228.4 mg/g d.w, the same concentration of glucose 59.2 mg/g d.w, similar to the minimum concentration reported in our range (59.73 mg/d d.w in VM), and a fructose content (72.5 mg/g d.w in *Prunus armeniaca* L. var. Tilton) higher compared our results. Finally, Akin et al. [46] showed a lower concentration compared to our results; in fact, the total sugar content ranged from 68.61 mg/d.w to 93.88 mg/g d.w and this could be derived from different cultivars and different regions.

Regarding the sourness of fruits, it depends largely on the relative amounts of sugars and organic acids present [47–50] and they are influenced by genotype, geographical

location of cultivation, and fruit stage of maturity [27,44,46,51]. Table 5 shows the organic acid content in Vesuvian apricots analyzed.

**Table 5.** Organic acid content (g/kg d.w) of the 12 accessions of Vesuvian apricot.

| | Ascorbic Acid | Citric Acid | Fumaric Acid | Malic Acid | Shikimic Acid | Succinic Acid | Tartaric Acid | Total Acids |
|---|---|---|---|---|---|---|---|---|
| | | | | **g/kg d.w** | | | | |
| BE | 0.78 ± 0.04 ab | 56.25 ± 2.27 a | 0.66 ± 0.00 fg | 10.37 ± 0.41 cde | 0.04 ± 0.01 de | 0.60 ± 0.09 d | 10.65 ± 0.84 cd | 79.35 ± 2.59 b |
| FR | 0.64 ± 0.07 d | 35.88 ± 1.67 b | 0.66 ± 0.00 fg | 11.03 ± 2.44 cde | 0.05 ± 0.00 cd | 0.63 ± 0.01 d | 20.54 ± 1.99 a | 69.42 ± 6.18 c |
| PAL | 0.40 ± 0.01 e | 34.77 ± 0.21 bc | 0.68 ± 0.00 de | 39.01 ± 3.12 a | 0.07 ± 0.00 b | 0.68 ± 0.02 d | 21.18 ± 0.83 a | 96.79 ± 2.06 a |
| PAN | 0.75 ± 0.01 bc | 19.06 ± 0.02 g | 0.65 ± 0.00 g | 10.27 ± 0.51 cde | 0.09 ± 0.00 a | 1.40 ± 0.10 bc | 19.45 ± 1.52 a | 51.66 ± 0.92 de |
| PAO | 0.81 ± 0.00 ab | 19.89 ± 0.63 fg | 0.71 ± 0.00 b | 11.26 ± 0.08 cd | 0.04 ± 0.01 e | 1.84 ± 0.57 b | 12.22 ± 1.69 c | 46.76 ± 1.69 ef |
| PE | 0.68 ± 0.04 cd | 32.90 ± 0.47 bcd | 0.72 ± 0.01 ab | 7.88 ± 0.39 e | 0.04 ± 0.00 e | 1.30 ± 0.09 c | 6.40 ± 0.02 e | 49.91 ± 1.02 de |
| PI | 0.73 ± 0.05 bcd | 34.57 ± 0.86 bc | 0.67 ± 0.00 f | 9.82 ± 0.19 cde | 0.00 ± 0.00 f | 0.56 ± 0.06 d | 8.81 ± 0.76 d | 55.51 ± 1.81 d |
| PO | 0.81 ± 0.09 ab | 35.58 ± 5.75 bc | 0.69 ± 0.01 c | 12.23 ± 1.66 c | 0.04 ± 0.00 e | 1.60 ± 0.18 bc | 1.75 ± 0.13 f | 52.71 ± 7.19 de |
| PU | 0.37 ± 0.03 e | 18.98 ± 0.55 g | 0.73 ± 0.00 a | 8.69 ± 0.22 de | 0.00 ± 0.00 f | 1.64 ± 0.10 bc | 9.30 ± 0.11 d | 39.71 ± 0.16 f |
| SC | 0.67 ± 0.01 cd | 24.03 ± 0.00 ef | 0.69 ± 0.00 cd | 11.17 ± 0.96 cd | 0.00 ± 0.00 f | 3.30 ± 0.17 a | 12.19 ± 0.20 c | 52.05 ± 0.94 de |
| VI | 0.86 ± 0.02 a | 30.74 ± 0.59 cd | 0.68 ± 0.00 e | 21.28 ± 1.24 b | 0.05 ± 0.00 c | 0.61 ± 0.02 d | 9.93 ± 0.88 cd | 64.14 ± 2.72 c |
| VM | 0.23 ± 0.01 f | 28.38 ± 3.33 de | 0.66 ± 0.00 fg | 9.47 ± 0.02 cde | 0.03 ± 0.00 e | 0.78 ± 0.10 d | 15.49 ± 1.28 b | 55.32 ± 4.53 d |
| Means | 0.64 ± 0.20 | 30.92 ± 10.13 | 0.68 ± 0.03 | 13.56 ± 8.48 | 0.04 ± 0.03 | 1.24 ± 0.79 | 12.32 ± 5.77 | 59.41 ± 15.48 |
| Significance | *** | *** | *** | *** | *** | *** | *** | *** |

All the data are expressed as the mean ± SD (standard deviation). The same letter indicates not significant differences according to Duncan's multiple range test ($p < 0.05$). Level of significance per the ANOVA is indicated as *** ($p < 0.001$).

The total organic acid content ranged from 39.71 g/kg d.w in 'PU' to 96.79 mg/kg d.w in 'PAL'; the latter showed the lowest pH value and the highest TA value (Table 2). Among organic acid citric, malic, fumaric, malic, ascorbic, shikimic, succinic and tartaric acid were found. Citric acid was the most abundant organic acid in the analyzed cultivars, with a concentration from 18.98 g/kg d.w in 'PU' to 56.25 g/kg d.w, in 'BE'. The other most concentrated organic acids were malic acid (from 7.88 g/kg d.w in 'PE' to 39.01 g/kg d.w in 'PAL'), an important sensory contributor for the sourness of fruits [26], and tartaric acid (from 1.75 g/kg d.w in 'PO' to 21.18 g/kg d.w in 'PAL'). Further, in 'PAL' the citric acid concentration was similar to the concentration of malic acid (34.77 g/kg d.w for citric acid and 39.01 g/Kg d.w for malic acid), while the citric acid content was similar to tartaric acid (19.06 g/Kg d.w for citric acid and 19.45 g/kg d.w for tartaric acid) in 'PAN'. It is reported that in many fruits, the increase in soluble sugar concentration during ripening complements the presence of malic and citric acids and imparts on the flesh a sweet taste [52]. Thus, in the unripe flesh of many fruits, a higher content of certain organic acids and a low content of sugars impart on the flesh a sour flavor, making them less palatable to animals [50]. The flavor plays an important factor in consumer choice and satisfaction, which results from a union of taste and aroma [26]. Additionally, Alajil et al. [50] and Karatas [25] showed that citric and malic acids were the most concentrated organic acids in apricots. The tartaric acid concentration was higher compared to Kim et al. [51], who showed a concentration range from 1.6 g/kg d.w to 3.0 g/kg d.w in Japanese apricots. On the contrary, Salur-Can et al. [52] showed that malic acid was more concentrated than citric acid (18 g/kg d.w for malic acid and 2 g/kg d.w for citric acid), while the authors showed a higher concentration (32 g/kg d.w) of succinic acid compared our results (from 0.59 g/kg d.w in 'BE' to 3.42 g/kg d.w in 'SC'). Additionally, Karabulut et al. [53] showed a higher concentration of malic acid, which ranged from 13.12 g/kg d.w to 18.77 g/kg d.w, than that of citric acid, which ranged from 14.59 mg/kg d.w to 25.03 mg/kg d.w, depending on apricot size. Furthermore, Su et al. [54] and Akin et al. [46] showed a variable content of both citric and malic acids that depended on apricot cultivar and region. The ascorbic acid content ranged from 0.23 g/kg d.w in 'VM' to 0.86 g/kg d.w in 'VI', and from 0.26 g/kg d.w to 0.35 g/kg d.w in the range reported by Akin et al. [46], who showed a concentration that ranged from 0.26 g/kg d.w to 0.97 g/kg d.w in different Turkic apricots. The ascorbic acid has an essential function in different cellular processes, such as cellular oxidation as well as to protect the cellular content against oxidative damage resulting from reactive oxygen species (ROS), which cause different types of chronic disease. Furthermore, ascorbic acid has antioxidant functions and antitumor and antiviral activities [55]. Further, our results showed a low and similar content among accessions. Finally, fumaric and shikimic

acids in all analyzed samples were present at a concentration lower than 1 g/kg d.w; also Hasib et al. [56], Bae et al. [57] and Schmitzer et al. [45] showed that fumaric and shikimic acids were the least concentrated organic acids in apricots. Malic and/or citric acid are abundant in the flesh of most fruits [48,57–60]. For example, their content can account for up to 40–50% of the dry weight of the unripe flesh of fruits such as apricot [61]. In many fruits, quinic acid can be abundant and can account for 20–30% of the dry weight of the flesh of some citrus fruits early in development [62]. In a smaller number of fruits, either oxalic or tartaric acid can be very abundant. The total amount of organic acids that are present in the fleshy parts of fruits of different species and their cultivars can vary greatly, and can also vary in terms of the relative abundance of the individual organic acids. In addition, these contents are dependent on both the tissue of the fruit and its stage of development, as well as on diverse environmental factors [48,59,60].

The content of organic acids and sugars significantly influences the sensory consistency, taste, and aroma of fruits [26,45]. Furthermore, from a nutritional point of view, organic acids contribute to maintain the acid–base balance in the intestine and to increase the bioavailability of iron [63]. Moreover, the organic acids can help to stabilize water-soluble vitamins B and C, as well as to stimulate appetite and digestion and absorption of minerals including potassium, copper, zinc, iron, and calcium. Additionally, due to their ability to chelate metals, the organic acids may act as antioxidants playing an important role in the protection of many diseases including cardiovascular diseases, cancer, and inflammation. Furthermore, the organic acids are involved in different biological processes. In fact, they reduce the growth of bacteria, which aids in the preservation of fruits [46]. Finally, the organic acid contents and their derivatives in the flesh of fruits have important dietary influences, affecting fruit taste and, in some cases, its suitability for processing into different fruit products [64–66].

### 3.3. Antioxidant Activity and Identification and Quantification of Vesuvian Apricot Bioactive

The current study aimed to provide useful data about the chemical composition and comprehensive evaluation of the antioxidant activity of extracts derived from 12 apricot Vesuvian accessions. As reported in the literature, a significant number of phenolic compounds were determined in the apricot extracts, although a significant variability was reported in terms of polyphenol content among the apricot accession samples [7].

Table 6 shows the total polyphenol content (TPC) detected with the Folin–Ciocalteu and the antioxidant activity evaluated with three different assays: DPPH, ABTS and FRAP.

**Table 6.** Total phenolic content (FOLIN) and antioxidant activity (DPPH, ABTS and FRAP) of the 12 Vesuvian apricot.

| Accessions | FOLIN (mg GAE/g d.w) | DPPH | ABTS mmol Trolox/kg d.w | FRAP |
|---|---|---|---|---|
| BE | 4.84 ± 0.24 b | 6.86 ± 0.42 ab | 15.83 ± 3.31 b | 15.01 ± 2.72 b |
| FR | 3.14 ± 0.50 e | 4.04 ± 0.59 efg | 12.22 ± 1.72 cd | 13.01 ± 1.50 c |
| PAL | 4.35 ± 0.19 c | 6.52 ± 0.88 bc | 13.96 ± 0.19 bc | 12.50 ± 1.20 c |
| PAN | 2.13 ± 0.07 hi | 1.83 ± 0.04 i | 6.65 ± 0.95 fg | 5.77 ± 0.10 ef |
| PAO | 2.00 ± 0.43 i | 2.43 ± 0.75 hi | 5.91 ± 0.91 g | 5.10 ± 0.95 f |
| PE | 3.66 ± 0.31 d | 5.57 ± 0.43 cd | 9.84 ± 0.66 de | 11.47 ± 0.84 c |
| PI | 2.86 ± 0.17 ef | 3.18 ± 0.44 gh | 8.48 ± 0.74 ef | 7.57 ± 0.95 de |
| PO | 5.69 ± 0.15 a | 7.58 ± 0.45 a | 19.77 ± 1.51 a | 19.69 ± 1.62 a |
| PU | 2.91 ± 0.10 ef | 5.83 ± 0.06 c | 11.17 ± 1.05 d | 8.52 ± 0.58 d |
| SC | 2.30 ± 0.27 ghi | 4.78 ± 0.52 de | 8.28 ± 0.67 efg | 6.72 ± 0.70 def |
| VI | 2.71 ± 0.34 i | 4.59 ± 0.35 def | 8.17 ± 0.45 efg | 7.21 ± 0.29 de |
| VM | 2.51 ± 0.50 fgh | 3.72 ± 1.41 fg | 10.48 ± 2.71 de | 6.52 ± 1.53 def |
| Means | 3.26 ± 0.27 | 4.74 ± 0.53 | 10.90 ± 1.24 | 9.92 ± 1.08 |
| Significance | *** | *** | *** | *** |

All the data are expressed as the mean ± SD (standard deviation). The same letter indicates not significant differences according to Duncan's multiple range test ($p < 0.05$). Level of significance per the ANOVA is indicated as *** ($p < 0.001$).

The antioxidant activity showed a high variability between the different accessions. 'PO' showed a higher polyphenol content than 'PAO' (+~184%), reporting values of 5.69 mg GAE/g d.w and 2.00 mg GAE/g d.w, respectively. 'BE', 'PAL' and 'PO' showed a higher polyphenol content than the general average of the Vesuvian apricot analyzed, equal to 3.26 mg GAE/g d.w. In all three antioxidant assays 'PO' reported statistically higher values than the other accessions. For DPPH it reported values of 7.58 mmol trolox/kg d.w approximately 60% higher than the general average recorded which was equal to 4.74 mmol trolox/kg d.w. Statistically lower values were instead recorded in the PAN accession with 1.83 mmol trolox/kg d.w. In the ABTS and FRAP assays, the 'PO' accession reported antioxidant activity values on average 81% higher compared to the general averagwhile 'PAO' was the accession that reported lower values, respectively, equal to 5.91 mol trolox/kg d.w and 5.10 83 mmol trolox/kg d.w. Among the cultivars analyzed, the variability of the antioxidant capacity obtained was consistent and this variability was in line with literature data. In fact, it is widely reported that the antioxidant activity varies according to factors such as apricot cultivar, geographical source, irrigation regimes and sample extraction protocols [36].

Our results have also shown a highly significant positive relationship between the total phenolic content and antioxidant activity (R = 0.76, 0.92 and 0.89 for DPPH, FRAP and ABTS, respectively). Such high R value suggested that the radical scavenging activity and ferric reducing ability could be credibly predicted on the basis of total phenolic content and directly confirmed that the phenolic compounds in the 12 Vesuvian apricot accessions were responsible for their antioxidant capacity. The present high correlation between antioxidant activity and phenolics agreed with previous studies [10].

The influence of different accession on the qualitative and quantitative profile of polyphenolic compounds of apricot fruits is included in Table 7.

**Table 7.** Quantitation of the main polyphenols in the investigated 12 Vesuvian apricot extracts. Results are expressed as the mean mg/100 g d.w. ± SD from three independent determinations.

| mg/100 g d.w | BE | FR | PAL | PAN | PAO | PE | PI |
|---|---|---|---|---|---|---|---|
| Pinoresinol | 0.15 ± 0.01 c | 0.15 ± 0.01 c | 0.34 ± 0.04 a | 0.10 ± 0.00 d | 0.09 ± 0.01 de | 0.07 ± 0.01 e | 0.21 ± 0.01 b |
| Quinic Acid | 30.07 ± 0.85 | 14.76 ± 0.42 d | 8.52 ± 0.28 de | 8.95 ± 1.47 de | 36.60 ± 1.07 b | 57.88 ± 3.56 a | 6.83 ± 0.60 e |
| Chlorogenic Acid | 65.11 ± 2.96 ab | 53.02 ± 1.33 c | 50.54 ± 8.40 cd | 30.81 ± 2.97 h | 23.60 ± 2.72 i | 36.17 ± 2.55 gh | 44.19 ± 5.41 ef |
| *p*-coumaric | 0.05 ± 0.02 e | 0.02 ± 0.01 e | 0.11 ± 0.05 cd | 0.12 ± 0.02 cd | 0.02 ± 0.01 e | 0.05 ± 0.03 e | 0.11 ± 0.02 cd |
| Caffeic Acid | 15.78 ± 4.61 abc | 13.78 ± 3.56 abc | 17.60 ± 3.87 a | 11.65 ± 5.38 abcd | 9.88 ± 1.92 cd | 10.21 ± 1.05 bcd | 16.55 ± 2.17 ab |
| Catechin | 14.50 ± 4.00 a | 8.02 ± 1.11 b | 1.28 ± 0.19 e | 1.27 ± 0.33 e | 2.46 ± 0.57 de | 6.92 ± 1.18 bc | 7.84 ± 0.90 b |
| Siringic Acid | 1.70 ± 0.24 bc | 1.30 ± 0.26 cd | 0.51 ± 0.56 ef | 0.37 ± 0.09 ef | 0.43 ± 0.16 ef | 0.03 ± 0.03 f | 2.20 ± 0.30 b |
| Epicatechin | 7.30 ± 0.75 bc | 9.37 ± 0.22 b | 7.50 ± 0.24 bc | 1.19 ± 0.28 d | 1.06 ± 0.07 d | 4.37 ± 0.14 cd | 8.26 ± 0.39 bc |
| Ferulic Acid | 3.49 ± 0.14 a | 2.39 ± 0.29 def | 2.57 ± 0.17 cdef | 2.75 ± 0.17 cd | 2.48 ± 0.21 def | 2.88 ± 0.54 cd | 2.17 ± 0.14 f |
| Naringin | 0.07 ± 0.02 de | 0.12 ± 0.03 bc | 0.06 ± 0.02 de | 0.06 ± 0.04 de | 0.06 ± 0.04 de | 0.14 ± 0.03 bc | 0.15 ± 0.05 b |
| Rutin | 75.21 ± 2.06 b | 82.07 ± 35.81 b | 140.58 ± 23.95 a | 48.79 ± 1.98 bc | 46.14 ± 53.28 bc | 75.44 ± 9.47 b | 75.37 ± 7.97 b |
| Quercetin-3-glucoside | 0.19 ± 0.04 cd | 0.22 ± 0.07 bc | 0.22 ± 0.02 bc | 0.20 ± 0.07 c | 0.19 ± 0.05 cd | 0.18 ± 0.02 cd | 0.22 ± 0.02 bc |
| Kaempferol-3-*O*-glucoside | 0.06 ± 0.02 bc | 0.06 ± 0.03 bc | 0.06 ± 0.02 bc | 0.06 ± 0.03 bc | 0.07 ± 0.04 b | 0.10 ± 0.01 a | 0.04 ± 0.01 c |
| Isorhamnetin-3-rutinoside | 0.45 ± 0.05 bc | 0.46 ± 0.07 bc | 0.32 ± 0.05 d | 0.53 ± 0.07 b | 0.34 ± 0.13 cd | 0.51 ± 0.06 b | 0.51 ± 0.04 b |
| Quercetin | 0.01 ± 0.00 c | 0.00 ± 0.00 c | 0.00 ± 0.00 c | 0.00 ± 0.00 c | 0.01 ± 0.01 bc | 0.02 ± 0.00 b | 0.00 ± 0.00 c |
| Luteolin-7-glucoside | 0.05 ± 0.01 bc | 0.04 ± 0.02 bc | 0.05 ± 0.01 bc | 0.05 ± 0.02 bc | 0.05 ± 0.03 bc | 0.06 ± 0.03 b | 0.03 ± 0.01 c |
| Myricitrin | 0.04 ± 0.00 bcd | 0.04 ± 0.01 abc | 0.04 ± 0.00 abc | 0.04 ± 0.01 cde | 0.04 ± 0.01 bc | 0.03 ± 0.01 de | 0.05 ± 0.01 ab |
| Total polyphenols | 214.22 ± 9.04 abc | 185.84 ± 31.35 cd | 230.32 ± 30.72 ab | 106.94 ± 4.66 f | 123.54 ± 49.53 f | 195.08 ± 14.96 bcd | 164.74 ± 7.73 de |

| mg/100 g d.w | PO | PU | SC | VI | VM | Means | Significance |
|---|---|---|---|---|---|---|---|
| Pinoresinol | 0.15 ± 0.01 c | 0.07 ± 0.01 e | 0.015 ± 0.02 c | 0.13 ± 0.01 c | 0.19 ± 0.03 b | 0.15 ± 0.08 | *** |
| Quinic Acid | 40.96 ± 0.68 b | 9.51 ± 0.30 de | 11.67 ± 1.50 de | 38.96 ± 12.98 b | 12.75 ± 3.43 de | 23.12 ± 16.83 | *** |
| Chlorogenic Acid | 43.97 ± 0.47 ef | 45.93 ± 5.42 de | 68.99 ± 0.67 a | 60.93 ± 2.34 b | 39.21 ± 4.94 fg | 46.87 ± 13.70 | *** |
| *p*-coumaric | 0.09 ± 0.02 d | 0.14 ± 0.02 c | 0.30 ± 0.04 a | 0.02 ± 0.01 e | 0.25 ± 0.02 b | 0.11 ± 0.05 | *** |
| Caffeic Acid | 13.23 ± 3.40 abcd | 16.94 ± 4.53 a | 14.16 ± 2.39 abc | 7.06 ± 3.31 d | 6.99 ± 7.44 d | 12.82 ± 5.00 | * |
| Catechin | 12.13 ± 4.24 a | 8.94 ± 0.98 b | 4.71 ± 0.38 cd | 1.30 ± 0.54 e | 3.37 ± 1.82 de | 6.06 ± 4.58 | *** |
| Siringic Acid | 0.00 ± 0.00 f | 2.31 ± 0.49 b | 0.74 ± 0.09 ef | 3.96 ± 1.00 a | 0.95 ± 0.41 de | 1.20 ± 1.18 | *** |
| Epicatechin | 7.23 ± 0.33 bc | 6.95 ± 0.77 bc | 4.18 ± 0.41 cd | 1.53 ± 0.26 d | 16.56 ± 9.01 a | 6.29 ± 4.79 | *** |
| Ferulic Acid | 2.69 ± 0.43 cde | 2.99 ± 0.08 bc | 3.37 ± 0.16 ab | 2.21 ± 0.61 ef | 2.72 ± 0.16 cd | 2.72 ± 0.48 | *** |

**Table 7.** *Cont.*

| mg/100 g d.w | PO | PU | SC | VI | VM | Means | Significance |
|---|---|---|---|---|---|---|---|
| Naringin | 0.03 ± 0.02 e | 0.12 ± 0.02 bc | 0.09 ± 0.03 cd | 0.14 ± 0.03 b | 0.25 ± 0.04 a | 0.11 ± 0.06 | *** |
| Rutin | 127.32 ± 25.67 a | 47.75 ± 6.63 bc | 56.52 ± 3.11 bc | 132.71 ± 8.03 a | 25.57 ± 11.59 c | 77.79 ± 40.94 | *** |
| Quercetin-3-glucoside | 0.44 ± 0.06 a | 0.12 ± 0.03 d | 0.24 ± 0.05 bc | 0.29 ± 0.03 b | 0.12 ± 0.06 d | 0.22 ± 0.09 | *** |
| Kaempferol-3-*O*-glucoside | 0.12 ± 0.02 a | 0.04 ± 0.01 bc | 0.06 ± 0.01 bc | 0.05 ± 0.01 bc | 0.04 ± 0.01 c | 0.06 ± 0.03 | * |
| Isorhamnetin-3-rutinoside | 0.54 ± 0.08 b | 0.26 ± 0.04 d | 0.28 ± 0.02 d | 0.85 ± 0.18 a | 0.31 ± 0.04 d | 0.45 ± 0.17 | *** |
| Quercetin | 0.09 ± 0.04 a | 0.00 ± 0.00 c | 0.00 ± 0.00 c | 0.00 ± 0.00 c | 0.01 ± 0.01 bc | 0.01 ± 0.00 | *** |
| Luteolin-7-glucoside | 0.09 ± 0.03 a | 0.03 ± 0.01 c | 0.04 ± 0.01 bc | 0.04 ± 0.01 bc | 0.02 ± 0.01 c | 0.05 ± 0.02 | ** |
| Myricitrin | 0.03 ± 0.01 e | 0.05 ± 0.01 abc | 0.06 ± 0.00 a | 0.04 ± 0.01 cd | 0.06 ± 0.01 a | 0.04 ± 0.01 | *** |
| Total polyphenols | 249.15 ± 28.32 a | 142.14 ± 3.29 ef | 165.55 ± 3.05 de | 250.22 ± 19.10 a | 109.36 ± 31.49 f | 178.09 ± 19.43 | *** |

All the data are expressed as the mean ± SD (standard deviation). The same letter indicates not significant differences according to Duncan's multiple range test ($p < 0.05$). Level of significance per the ANOVA is indicated as * ($p < 0.05$), ** ($p < 0.01$), *** ($p < 0.001$).

Polyphenols have attracted more attention in recent years because of their antioxidant capacity useful to prevent several chronic diseases [67]. Apricot occupies a distinct position among stone fruits due to its multifaceted compositional contour and significant functional potentials. Apricots contain a wide variety of phytochemicals that function as antioxidant. Literature reports that apricots contain appreciable amounts of total phenolic compounds and flavonoids which make them more valuable as functional food. The most representative polyphenols are quinic acid, chlorogenic acid, caffeic acid, epicatechin, rutin. Quinic acid showed high significant values in the 'PE' accession with values of 57.88 mg/100 g, and lower values were shown in the 'PI' accession with 6.83 mg/100 g. Chlorogenic acid showed significant values in the 'SC' accession with values of 68.99 mg/100 g, and the lowest values have been reported in the 'PAO' accession with 23.60 mg/100 g. Caffeic acid showed significantly high values in accession 'PAL' (17.60 mg/100 g) and 'PU' (16.94 mg/100 g), and less values in accession 'VI' (7.06 mg/100 g) and 'VM' (6.99 mg/100 g). Epicatechin values were high in the 'VM' accession with values of 16.56 mg/100 g, compared to 'PAN', 'PAO', 'VI' with values of 1.19, 1.06, 1.53 mg/100 g, respectively. Rutin, instead, was the most significant polyphenolic compound among all present with an average value of 77.79 mg/100 g, and its values were highly significant in the 'PAL', 'PO', 'VI' accessions with high values of 140.58, 127.32, 132.71 mg/100 g, respectively, less significant instead in the 'VM' accession with 25.57 mg/100 g. 'PO' and 'VI' showed a total polyphenol content approximately 29% higher than the general average, reporting values of 249.15 mg/100 g and 250.22 mg/100 g, respectively. The 'PAN' accession, on the other hand, reported a lower quantity than the general average of approximately 65% with values of 106.94 mg/100 g. Radi et al. [68], characterized and identified the main phenolic compounds in French apricot. Eight compounds were isolated and identified by comparing their characteristics with commercial standards, namely protocatechuic acid, (+)-catechin, chlorogenic acid, (-)-epicatechin, naringenin-7-glucoside, quercetin-3-glucoside, quercetin-3-rhamnoglucoside also namely rutin, and kaempferol-3-rutinoside. The obtained results show that chlorogenic and neochlorogenic acids, (+)-catechin, (-)-epicatechin and rutin are the major compounds among the polyphenols. Di Vaio et al. [22], evaluated the primary quality characteristics, including total antioxidant activity and total polyphenol content, of the most popular autochthonous apricot cultivars in Campania: Ceccona, Vitillo and Pellecchiella. Pellecchiella exhibited the highest Trolox Equivalent antioxidant capacity, whereas Ceccona had the largest concentration of phenolic compounds. The highest concentration of total phenolic compounds was found in Ceccona, which had 389.93 μg/g FW, followed by Vitillo (184.36 μg/g FW) and Pellecchiella (131.35 μg/g FW). In accordance with our previous work, the results obtained showed that the cultivar "Vitillo" has a higher concentration of the two most represented compounds (chlorogenic acid and rutin) than the cultivar "Pellecchiella. Göttingerová et al. [69], identified phenolic compounds, organic acids, vitamin C, flavonoids, antioxidant capacity, and carotenoids in the fruits of particular 15 Czech apricot cultivars in order to demonstrate the high nutritional value of apricot. Chlorogenic acid is among the primary phenolic acids found in the fruit of the Prunus genus in the concentration range of 0.69 to 21.94 mg/100 g

FW. The most prevalent flavanols from the flavan-3-ol group are catechin and epicatechin, which is present in a variety of plant species. Catechin was the most prevalent of these substances, with values ranging from 0.55 to 10.75 mg/100 g FW. The total polyphenol content in the set of cultivars under investigation ranged from 57.33 to 571.93 mg GAE/100 g FW. The antioxidant activity measured by using the assay was in the average of 203.42 mg TE/100 g FW. Antioxidant properties of fruit depend on the cultivar and are related to climatic conditions. Cultivar, growing region, and ripening stage play an important role in apricot's nutritional and quality qualities. The harvest of fruits represents a crucial step to obtain the optimal compositions of bioactive compounds to determine the ideal time to harvest. For this reason, is important the chemical characterization of cultivars [70]. In addition, as previously reported [22] the quali-quantitative variability in the polyphenols composition of apricot extracts may be also caused by the genetic characteristics of the examined accessions. Particularly, the amounts of polyphenols produced during ripening could depend on the influence of affect biosynthetic pathways. Determining unambiguous fruit quality criteria has always been extremely challenging due to the genetic heterogeneity of apricot varieties. This study emphasizes the requirement to evaluate the apricot fruit's possible health advantages. Due to its pharmacologically significant bioactive components, it has been reported having beneficial in treating conditions such as chronic gastritis, oxidative intestinal damage, hepatic steatosis, coronary heart disease and atherosclerosis [63]. To create a better understanding of foods' roles and functions in various disease prevention processes, it is necessary to pose more attention to foods and each of their constituent components. In-depth research will aid in making use of this resource in several human diseases.

*3.4. Principal Component Analysis (PCA)*

To provide a summary of the characteristic of qualitative parameters of the Vesuvian apricot a principal component analysis (PCA) was carried out, which separated the accessions based on them qualitative traits. The principal components (PCs) (Figure 2) explained 69.27% of the total variance. PC1 explained 45.74% of the total variance and was positively correlated with a higher antioxidant activity and total polyphenol content. PC1 was negatively correlated by TA and total sugar content. The second principal component (PC2) explained 23.53% of the variance and was correlated with a higher pH and total sugar content while being negatively correlated with antioxidant acidity, TA and total acids. 'PAN' and 'PE' showed a positive correlation with the total sugar content while these parameters are negatively correlated with the 'BE', 'PI', and 'VM' accessions. The latter instead showed a positive correlation with all three assays of antioxidant activity (DPPH, FRAP, and ABTS) and total acids. 'FR', 'PAO' and 'SC' showed a positive correlation with pH, TSS and total polyphenols. Instead, these parameters were negatively correlated to the 'PO' and 'PAL accessions, which showed a greater correlation with TA.

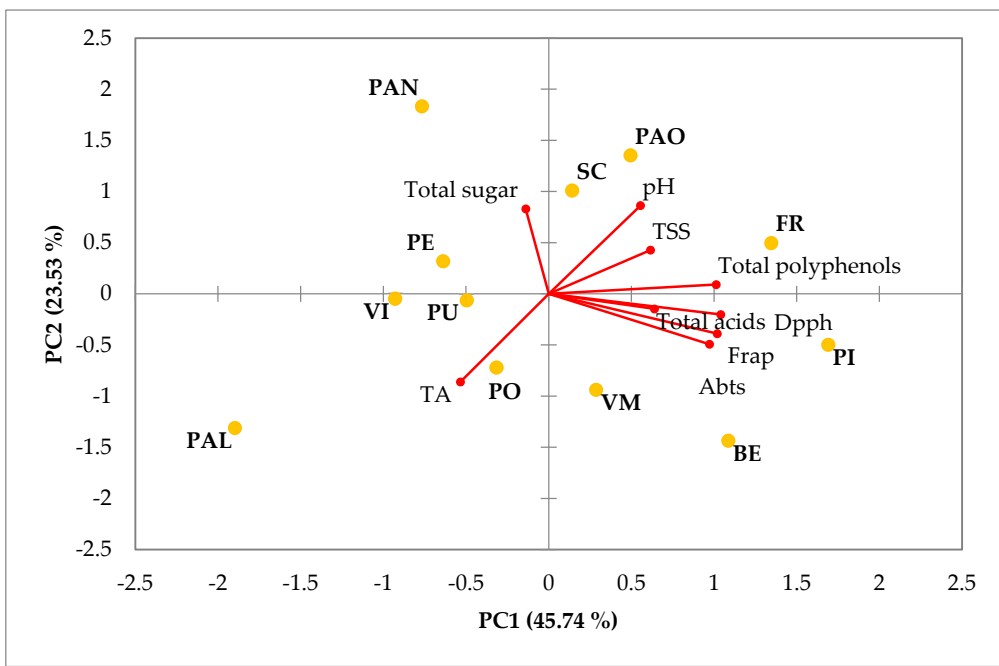

**Figure 2.** Principal component analysis (PCA) of qualitative and bioactive parameters of Vesuvian apricots: TSS, TA, pH, total sugar content, total acid content and total polyphenols.

## 4. Conclusions

Based on our results, the Vesuvian apricots showed significant differences in all parameters analyzed. The TSS showed a range from 18.17 °brix for 'SC' to 13.33 °brix for 'VI', and lower acidity values, 6.90 g/L citric acid, in the 'PAO' accession, and higher values, 22.17 g/L malic acid, in 'PAL'. As regards the bioactive composition of the fruits, the accessions 'PO' and 'BE' exhibited the highest antioxidant activity in all three antioxidant assays, while 'PO' showed the highest total phenolic content, 249.15 mg/100 g dw. These cultivars can be selected to promote their positive effect on health or in breeding programs to improve this parameter of new varieties.

Moreover, selecting and cultivating fruit varieties with a higher content of bioactive molecules is an excellent way to encourage agricultural sustainability, and integrating them into the diet can improve health benefits for consumers. The results obtained suggest that the Vesuvian apricots present fruits of superior quality.

**Author Contributions:** Conceptualization, C.D.V., A.R. and R.R.; methodology, G.G., L.D.L., L.I. and A.C.; software, A.C.; validation, C.D.V., A.R. and R.R.; formal analysis, M.C. and A.C.; investigation, L.D.L., G.G., L.I. and A.C.; resources, C.D.V., R.R. and A.R.; data curation, A.C., M.C. and L.D.L.; writing—original draft preparation, A.C., L.D.L., L.I. and G.G.; writing—review and editing, L.I., G.G., L.D.L. and A.C.; supervision, C.D.V., A.R. and R.R. All authors have read and agreed to the published version of the manuscript.

**Funding:** This study was carried out within the Agritech National Research Center and received funding from the European Union Next-Generation EU (PIANO NAZIONALE DI RIPRESA E RE-SILIENZA (PNRR)—MISSIONE 4 COMPONENTE 2, INVESTIMENTO 1.4—D.D. 1032 17/06/2022, CN00000022). This manuscript reflects only the authors' views and opinions; neither the European Union nor the European Commission can be considered responsible for them. Furthermore, this research was funded by PSR Campania 2014/2020 Research Project Measure 10—Type of Intervention 10.2.1—Conservation of indigenous genetic resources to protect biodiversity—Vegetable Genetic Resources—Project "DiCoVaLe: Diversity, Conservation and Enhancement of Woody species of native fruit from Campania CUP: B24I19000440009" National Project: DICA n°2019.0601893, 8 October 2019.

**Data Availability Statement:** Not applicable.

**Acknowledgments:** The authors are grateful to Sara Sepe for her technical assistance in the laboratory analysis. The authors thank the "Fresystem S.p.A." and "Cupiello" for providing kind support to the Conservation Center of Campania Fruit Biodiversity.

**Conflicts of Interest:** The authors declare no conflict of interest.

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
