# Peer review of "Biochemical and Nutraceutical Characterization of Different Accessions of the Apricot (Prunus armeniaca L.)"

_horticulturae, doi:10.3390/horticulturae9050546_

Round 1

Reviewer 1 Report

This work represents a lot of dedication and shows us very interesting information

Could be recommended for publication with minor corrections.

Observations

The work is very descriptive although repetitive, it is not enough to report that the value is higher or lower than the results previously published by other authors. I suggest more discussion.

It is essential to integrate the relationship of the study with the biochemical and nutraceutical characteristics. I cannot find what the nutraceutical relationship is? what are the components that could be related to nutrition or a beneficial aspect for human health?

The PCA analysis should give us information on the relationships between the analyzed components, I suggest a better analysis and a specific conclusion.

What is the general objective? two or three different objectives are mentioned throughout the paper.

Regarding the organoleptic characterization, I cannot find what are the chemical components related to the organoleptic characteristics?. That result is not clear.

In general, I refer to relationship structure-function which I believe is important to mention, at least to suggest the benefits to human health and clarify the compounds related whit the quality characteristics of the fruits.

I think that all the cultivars analyzed have beneficial characteristics whit a greater or lesser degree depending on their composition.

It is convenient to mention the culture conditions and how the environment could affect the biosynthesis of biosynthesis pathways.

Author Response

REVIEWER 1

This work represents a lot of dedication and shows us very interesting information.

Could be recommended for publication with minor corrections.

Observations

The work is very descriptive although repetitive, it is not enough to report that the value is higher or lower than the results previously published by other authors. I suggest more discussion.

  1. It is essential to integrate the relationship of the study with the biochemical and nutraceutical characteristics. I cannot find what the nutraceutical relationship is? what are the components that could be related to nutrition or a beneficial aspect for human health?
  2. Dear Reviewer, Thank you for your suggestion. We have improved discussions, and more information has been added at lines 541-550, 616-622, 683-691.

Lines 541-550 : “The content of organic acids and sugars significantly influences the sensory consistency, taste, and aroma of fruits [26,45]. Furthermore, from a nutritional point of view, organic acids contribute to maintain acid base balance in the intestine and to increase bioavailability of iron [63]. Moreover, the organic acids can help to stabilise water-soluble vitamins B and C, as well as to stimulate appetite and digestion and absorption of minerals including potassium, copper, zinc, iron, and calcium. Additionally, due to their ability to chelate metals, the organic acids may act as antioxidants playing an important role on the protection of many diseases including cardiovascular diseases, cancer, and inflammation. Furthermore, the organic acids are involved in different bio-logical processes. In fact, they reduce the growth of bacteria, which aids in the preservation of fruits [46]”

Lines 616-622: “Polyphenols have attracted more attention in recent years because of their antioxidant capacity useful to prevent several chronic diseases [67]  . Apricot occupies a distinct position among stone fruits due to its multifaceted compositional contour and significant functional potentials. Apricots contain a wide variety of phytochemicals that function as antioxidant. Literature reports that apricots contain appreciable amounts of total phenolic compounds and flavonoids which make them more valuable as functional food.”

Lines 683-691: “This study emphasizes the requirement to evaluate the apricot fruit's possible health advantages. Due to its pharmacologically significant bioactive components, it has been reported having beneficial in treating conditions like chronic gastritis, oxidative intestinal damage, hepatic steatosis, coronary heart disease and atherosclerosis [63]. To create a better understanding of foods' roles and functions in various disease prevention processes, it is necessary to pose more attention to foods and each of their constituent components. In-depth research will aid in making use of this resource in several human diseases.”

  1. The PCA analysis should give us information on the relationships between the analyzed components, I suggest a better analysis and a specific conclusion.

2.We have already done other analyzes before (cluster analysis) but we think the PCA can be the one that best summarizes the results.

  1. What is the general objective? two or three different objectives are mentioned throughout the paper.
  2. The specific objective is a varietal characterization of the Campania cultivars, with attention to the biochemical and nutraceutical characteristics.
  3. Regarding the organoleptic characterization, I cannot find what are the chemical components related to the organoleptic characteristics?That result is not clear.
  4. The results have been implemented, and more information have been added at lines 441-444.

“Regarding the sourness of fruits, it depends largely on the relative amounts of sugars and organic acids present [47-50,48] and they are influenced by genotype, geographical location of cultivation and fruits stage of maturity [27,44,46, 51]. Table 6 shows the organic acids content in Vesuvian apricots analyzed”.

  1. In general, I refer to relationship structure-function which I believe is important to mention, at least to suggest the benefits to human health and clarify the compounds related whit the quality characteristics of the fruits. I think that all the cultivars analyzed have beneficial characteristics whit a greater or lesser degree depending on their composition. It is convenient to mention the culture conditions and how the environment could affect the biosynthesis of biosynthesis pathways.
  2. This aspect has been implemented in different paragraphs of manuscript.

Reviewer 2 Report

The study is very interesting and well carried out. The document is generally well written but the following needs to be improved: Much of the bibliography needs to be updated. Differences between accessions should be discussed further. I include more suggestions in the pdf. I consider that the manuscript can be accepted

Author Response

The study is very interesting and well carried out. The document is generally well written but the following needs to be improved: Much of the bibliography needs to be updated. Differences between accessions should be discussed further. I include more suggestions in the pdf. I consider that the manuscript can be accepted.

  1. Line 32- Statistical data must have very recent references

The reference has been updated: Rakida, A. Analysis of morphological and pomological features of apricot in the Nakhchivan Autonomous Republic of Azerbaijan. Turk J Agric For.2023 47(1), 23-30.

Line 36 - Statistical data must have very recent references

The reference has been updated: Rampáčková, E., Mrázová, M., Čížková, J., & Nečas, T. Pomological Traits and Genome Size of Prunus armeniaca L. Considering to Geographical Origin. Acta Hortic, 2022, 8(3), 199.

Line 38 - reference appropriately

This reference is correct

Line 41 - to include reference

This reference has been added: Basile, B., Mataffo, A., Forlani, M., & Corrado, G. Diversity in Morphometric, Pomological, and Fruit-Quality Traits of Apricot (Prunus armeniaca) Traditional Varieties: Implications for Landrace Differentiation at Regional Scale. Diversity, 2022, 14(8), 608.

Line 44 - reference more recent

The reference has been updated: Basile, B., Mataffo, A., Forlani, M., & Corrado, G. Diversity in Morphometric, Pomological, and Fruit-Quality Traits of Apricot (Prunus armeniaca) Traditional Varieties: Implications for Landrace Differentiation at Regional Scale. Diversity, 2022, 14(8), 608.

Line 59 - to include reference

The reference has been updated: Lo Bianco, R.; Farina, V.; Indelicato, S.G.; Filizzola, F.; Agozzino, P. Fruit Physical, Chemical and Aromatic Attributes of Early, Intermediate and Late Apricot Cultivars: Fruit Quality and Ripening Season in Apricot. J. Sci. Food Agric. 2010, 90, 1008–1019, doi:10.1002/jsfa.3910.

Line 69 - reference more recent

The reference has been updated: Modi, B., Timilsina, H., Bhandari, S., Achhami, A., Pakka, S., Shrestha, P., ... & Parajuli, N. Current trends of food analysis, safety, and packaging. Int. J. Food Sci, 2021.           

Line 72 - reference more recent

The reference has been updated: Akhone, M. A., Bains, A., Tosif, M. M., Chawla, P., Fogarasi, M., & Fogarasi, S. Apricot kernel: bioactivity, characterization, applications, and health attributes. Foods, 2022, 11(15), 2184.

Line 91 - the introduction is well covered, it includes enough information to put the reader in context, the problem and the objective are clear, but most of the citations are outdated.

Dear Reviewer, Thank you for your suggestions. Where possible, in the introduction the bibliography has been updated and replaced.

  1. Line 55 - reference is 2009, this statement is still valid?

Yes, it is valid.

  1. Line 118 - What is the criterion to indicate that they are ripe for consumption? color? include more details.

More details have been added at line 119. Each sample consisted of 15 apricot fruits harvested at consumer maturity, which showed a pulp firmness quite low, in fact, apricot softening is a crucial parameter along the whole post-harvest supply chain, one that is highly influenced by the cultivar, storage conditions and duration [23].

  1. Line 126 – Delete

It was deleted.

  1. Line 253 - experimental design? randomized?

This information has been added to the line 103.

  1. Line 254 - to include assumptions of normality and homogenity of variances

The Shapiro Wilk’s was used to assess the normality of the datasets.

  1. Line 285 - and PE

Done

  1. Line 291 - It is important to explain what the differences in the weight of the accessions are due to.

This difference is due to the variability of the different genotypes analyzed, this information has been added at line 338. “The ANOVA test reported significant statistical differences between the weight of the various accessions which reported an average weight of about 55.96 g, this difference is due to the variability of the different genotypes analyzed, all grown under the same environmental conditions.”

  1. Line 346 – space

Done

  1. Line 354 – delete

Done

  1. Line 364 - this paragraph must go before the table

Done

  1. Line 371-376 I consider that there is a lot of information from other studies, better indicate if the values obtained are lower, higher or similar to those reported in other studies

This part has been modified. “The content of both sucrose and glucose was higher compared to the range reported by Naryal et al. [42] that showed a range between 67.9 mg/g to 366.5 mg/g, 37.3 mg/g to 152.9 mg/g for sucrose and glucose, respectively, while the fructose content was lower for BE, PAN, PI, PO, VI and MI respect the minimum content of range reported by Naryal et al. [42] that showed a range content of 27.6 mg/g to 155.3 mg/g, depending on the grown localities of fruits”.

  1. Line 386 - but the study was done on fruits in a single stage of maturity, right?

Yes, the study was carried out in the same stage for all accession, at consumer stage of maturity.

  1. Line 401- table no include means and significance.

They have been added.

  1. Line 414 - reference?

This reference has been added.

  1. Line 480 - to include this test in statistical analysis.

This information have been added at line 304.

  1. Line 481 - only R, because R2 is coefficient of determination.

Done

  1. Line 504 - this paragraph is too long

The paragraph has been modified.

  1. Line 513 – Delete

Done

  1. Line 570 - The description of the results seems appropriate to me. The discussion is mostly focused on comparing with other studies, even the comparison in some variables is excessive, I suggest reducing this aspect. And it remains to explain in more detail the differences between accessions, because this is the central theme of the manuscript.

The requested changes have been made in the paragraph.

  1. Delete

Done